# Influence of PEO Electrolyzer Geometry on Current Density Distribution and Resultant Coating Properties on Zr-1Nb Alloy

**DOI:** 10.3390/ma16093377

**Published:** 2023-04-26

**Authors:** Veta Aubakirova, Dmitry Gunderov, Ruzil Farrakhov, Vasily Astanin, Andrey Stotskiy, Arseny Sharipov, Alexey Demin, Leonard Khalilov, Evgeny Parfenov

**Affiliations:** 1Department of Electronic Engineering, Ufa University of Science and Technology, 12 Karl Marx Street, Ufa 450008, Russia; veta_mr@mail.ru (V.A.); frg1982@mail.ru (R.F.); v.astanin@gmail.com (V.A.); stockii_andrei@mail.ru (A.S.); arsenyarseny36728@gmail.com (A.S.);; 2Institute of Molecule and Crystal Physics, Ufa Federal Research Center, Russian Academy of Sciences, 151, Prospekt Oktyabrya, Ufa 450075, Russia; 3Department of Materials Science and Physics of Metals, Ufa University of Science and Technology, 12 Karl Marx Street, Ufa 450008, Russia; evparfenov@mail.ru; 4Institute of Petrochemistry and Catalysis, Ufa Federal Research Center, Russian Academy of Sciences, 141, Prospekt Oktyabrya, Ufa 450075, Russia; khalilovlm@gmail.com

**Keywords:** zirconium alloy, plasma electrolytic oxidation, coating morphology, computer simulation of electric field

## Abstract

This paper is devoted to the study of the current density distribution effect on plasma electrolytic oxidation process and resultant coatings on a Zr-1Nb alloy. The influence of the distance between the plates simultaneously placed into an electrolyzer was evaluated to assess the throwing power of the PEO process. The current density on the facing surfaces of the plates decreases when the distance between them shrinks. This current density has a notable impact on the resultant PEO coating in terms of the surface morphology parameters and electrochemically evaluated corrosion resistance. The influence of this effect is low on the stages of anodizing and spark discharges (60–120 s of the PEO), and significantly increases on the stage of microarc discharges (120–360 s of the PEO). The coating obtained with a smaller distance between the plates, while having the same coating thickness as the others, exhibits higher wear resistance. New correlations between the current density, diffusion coefficient, time constant of nucleation and the coating thickness in the middle of the facing samples were established; in addition, a correlation of the coating morphology in this area with the roughness parameters RPc, RSm was shown. This study contributes to the development of optimized PEO processes for the simultaneously coated several devices of complex shape, e.g., orthopedic implants.

## 1. Introduction

Plasma electrolytic oxidation (PEO) is an environmentally friendly surface treatment method that is successfully used to create wear-resistant, corrosion-resistant, heat-resistant, biocompatible and decorative coatings on aluminum, titanium, zirconium and other light alloys [1,2,3,4]. Unlike aluminum and titanium, zirconium alloys are less investigated in terms of the specific features of the PEO process mechanism; however, recent studies appear to be focused on Zr and its alloys [5,6,7]. One important Zr alloy application is nuclear fuel cladding which suffers from hydrogen embrittlement in the boiling water environment of a power plant [8,9]; the PEO coatings show a good promise for fuel cladding corrosion protection [10,11].

PEO process can produce biocompatible coatings on zirconium implants due to the porous morphology that imitate the bone surface [12,13,14]. For medical implants, it is crucial that the surface treatment procedure provides coatings of the required quality on devices that typically have complicated shapes, with threaded connections and holes. Moreover, industrial production requires simultaneous PEO treatment of several devices in one batch; recent research shows the necessity of developing a special approach for group plasma electrolytic oxidation [15].

Therefore, it is important to understand how the PEO process mechanism and regimes of the treatment should be amended to make a step from a simple-shaped sample to a load of real implants. This problem can be solved by considering a boundary value problem of current density distribution in the system with multiple samples. Recent studies investigate the influence of the current density distribution on the morphology of a PEO coating on complex-shaped parts using a magnesium alloy as an example [16]. There are not many publications regarding spatial current density distribution in plasma electrolytic treatments, because the system is non-linear, and this problem requires solving Laplace and Navier–Stokes’ partial differential equations [17,18].

The current density is one of the primary technological parameters of the PEO [19]. The impact of current density on the properties of PEO coatings has been the subject of numerous studies [20,21,22]. The distribution of the current density is non-uniform due to the complex-shaped treated device having edges and deep holes. The current density also varies as a result of changes in the distance between two or more nearby samples being at the same electric potential. It is challenging to predict the value of the current density distribution over the surface of a treated implant with a complex shape without a numerical model. Ansys, Comsol and Elcut are among the software products that are suitable for the electric field, charge and mass transfer simulation. The software uses the finite element method to solve the boundary value problem that was indicated above.

Therefore, in the previous studies, the authors identified a gap in assessment of the throwing power of the PEO process with respect to the multiple workpieces coated in one load. This gap was addressed with an example of the investigation of PEO coating properties on a zirconium alloy as a function of various current densities obtained at different spatial arrangements of the samples in one batch. Compared to other works [15,16,17,18], this study jointly considers both mathematical models of the electric field and the kinetics of the electrochemical reactions underlying the PEO coating process of the multiple workpieces. The PEO coating thickness, morphology, phase composition, corrosion resistance and wear resistance were evaluated as the key properties, and new correlations between the process parameters were established. This research opens the prospects for rational batch PEO treatments of implants in industrial manufacturing.

## 2. Materials and Methods

### 2.1. Plasma Electrolytic Oxidation Experimental Setup

To assess the influence of the spatial arrangements of the samples, two 15 × 15 mm zirconium alloy parallel plates were PEO treated. The anode potential was applied to both plates. As shown in Table 1, the plates were spaced apart by three different distances *d*: 2.5 ± 0.5 mm, 5 ± 0.5 mm and 10 ± 0.5 mm (experiments D1 to D3). In experiment D4, a single plate was PEO treated. This arrangement alters the current density distribution on the surfaces between the plates compared to the surfaces of the single sample D4 as a result of the electric field shielding. Figure 1 shows the area—a circle with a diameter of 5 ± 0.5 mm—on which measurements and SEM images were obtained.

### 2.2. Metal Sample Preparation

In this study, samples made of Zr-1Nb alloy were used; Table 2 lists the material chemical composition. The following steps were used to prepare the samples prior to the plasma electrolytic oxidation: the samples were polished with a SiC sandpaper P400 to P4000 grit, then ultrasonically cleaned sequentially in distilled water and isopropyl alcohol (each step for 5 min). A wire was inserted into a hole located at the sample corner in order to make a good electric contact; the wire contact was insulated with an epoxy resin.

### 2.3. PEO Coating Formation

The samples were PEO treated in a 10 L tank with the electrolyte having a temperature of 20 ± 1 °C. An aqueous solution of 15 g/L sodium phosphate, 25 g/L calcium acetate, 1 g/L sodium hydroxide and 1 g/L boric acid was selected as the electrolyte composition, as it previously demonstrated the best coating properties [14]. The electrolyte was stirred with an electromagnetic stirrer located under the tank. Using Anion-4100 (Anion, Novosibirsk, Russia), the electrolyte-specific conductivity was estimated experimentally as 14.96 ± 0.04 Cm/cm.

The pulsed unipolar mode was employed by the PEO equipment (USATU, Ufa, Russia) [23]. The pulse frequency was 700 Hz; the pulse voltage magnitude was 480 V. The pulses had a 26% duty cycle. The pulse voltage magnitude was increased for 120 s from 0 to 480 V, and then it was stabilized at that setpoint. The PEO coating formation time was 6 min. The coating was formed on the samples connected as the anode in the center of the electrolyte tank. The cathode was a stainless steel cooling coil that was positioned along the electrolyte tank walls. Figure 2 shows a top view of the cathode and anode arrangement.

### 2.4. Electric Field Modeling

The system was modeled in ELCUT software v. 6.3 (TOR, St. Petersburg, Russia) that solves the Laplace equation based on a static model of the electric field in conductive medium. The “Electric field of direct currents” problem type was chosen in the ELCUT software to analyze the field in the electrolyte during the PEO process. Figure 2 illustrates a grid of finite elements. “Cathode,” “Anode,” and “Electrolyte” block properties were set up; the electrolyte-specific conductivity was set as measured above. Potentials of 0 and 480 V were attributed to the “Cathode” and “Anode” as boundary values, respectively. Because the coating is approximately three times thinner than the sample, it was modelled with a thickness of 0.2 mm for visual purposes. The ratio of the magnitudes of the current and voltage pulses on the recorded waveforms, taking into account the electrolyte resistance and sample area, was used to determine the coating-specific conductivity, following the approach developed elsewhere [18].

### 2.5. Estimation of Kinetic Coefficients of the PEO Process

The method of electrochemical relaxation [24], which is based on the removal of the electrochemical system from an equilibrium using the voltage step and subsequent observation of current relaxation back to the equilibrium or a new stationary state, was used to study the kinetic coefficients describing the plasma electrolytic oxidation of the Zr-1Nb alloy. To quantify the growth kinetics, the pulse magnitudes of the anodic current density were studied in this work using the chronoamperometric technique. According to the method, the kinetic coefficients were examined in order to study the current transients in the relaxation zone. The supply voltage pulse served as the input step, and the output was the current density *j* magnitude values. This method was successfully used in another study of the kinetics of the PEO process of a magnesium alloy [25].

We applied the kinetic model that contains three terms: for the metal dissolution, for 3D crystallisation of the oxides, and for the oxygen evolution. The formula’s derivation is given elsewhere [24,26]:(1)$$j=j_{0}\exp(\frac{-t^{m}}{\tau })+zFc\sqrt{\frac{D}{\pi t}}\left(1-\exp(\frac{-t^{m}}{\tau })\right)+j_{C},$$
where *j*_0_—initial (maximum) current density magnitude; *m*—type of crystallisation; *z*—nuclear charge; *F*—Faraday’s constant; *c*—concentration of metal in the coating; *D*—diffusion coefficient; *τ*—time constant of nucleation; and *j*_C_—constant current density for the oxygen evolution. The transient plots of the anodic current density magnitude values were interpolated using the Curve Fitting Toolbox (Matlab); the coefficient of determination R^2^ was used to characterize the goodness of the fit.

### 2.6. Surface Characterization

A DeFelsko Positector 6000 eddy current probe with an N-type probe was used to measure the coating thickness. A stylus profilometer TR220 (Time Group, Beijing, China) was used to quantify the surface roughness along a track of 0.5 cm, in accordance with ISO 4287:2000 standard.

A scanning electron microscope (SEM), the Hitachi Regulus 8220 (Hitachi, Tokyo, Japan), was used to study the surface topography and coating microstructure. The micrographs taken with the scanning electron microscope were used to estimate the porosity of the coating using the ImageJ program (U. S. National Institutes of Health, Bethesda, MD, USA) as the percentage of the surface area taken by the pores; a corresponding histogram of the pore area distribution was also calculated.

The X-ray diffractometry phase analysis was carried out on the Bruker D2 Phaser (Bruker AXS GmbH, Karlsruhe, Germany). The X-ray tube provided CuKα radiation; a step scan from 25 to 80 degrees 2θ was realized. The resulting diffractograms were processed using the DIFFRAC.EVA v.5.2 software (Bruker AXS GmbH, Karlsruhe, Germany) with the PDF-2 2021 database (ICDD, Newtown Square, PA, USA).

### 2.7. Evaluation of the Electrochemical Properties of the Samples

The evaluation of the electrochemical properties of the samples was carried out in Ringer’s solution at 37 °C using R-5X electrochemical station (Elins, Moscow, Russia). According to the manufacturer’s information (Solofarm, St. Petersburg, Russia), 1 dm^3^ of Ringer’s solution contains NaCl (8.6 g), KCl (0.3 g) and CaCl_2_∙6H_2_O (0.25 g), dissolved in a distilled water. The potentiodynamic polarization (PDP) tests were carried out in the range of ±600 mV with respect to the open circuit potential (OCP), with a scan rate of 0.25 mV/s. A three electrode setup, with a graphite rod as a counter electrode and a silver chloride electrode (3.5 M KCl) as a reference, was used. The results of the potentiodynamic polarization were analyzed by Tafel’s method. The polarization resistance *R*p was calculated from the polarization curve range ±10 mV with respect to the free corrosion potential.

### 2.8. Wear Resistance Tests

The wear resistance properties of the coatings were investigated on a Nanovea tribometer (Nanovea Inc., Irvine, CA, USA); the sample was paired against an Al_2_O_3_ ball with a diameter of 6 mm at a constant sliding speed of 3000 mm/min under a load of 10 N. The aforementioned Ringer’s solution was used as a lubricant, 100 mL of which was poured into the tribometer cup before the tests. Five points on each sample were tested.

### 2.9. Correlation Analysis

A correlation analysis of the obtained data was carried out. The Pearson correlation coefficient was used [27]:$$r=\frac{\sum \left(X-\overset{-}{X})(Y-\overset{-}{Y}\right)}{\sqrt{\sum {\left(X-\overset{-}{X}\right)}^{2}\sum {\left(Y-\overset{-}{Y}\right)}^{2}}}\cdot 100\%$$
where *X* and *Y* are samples of the analyzed pair of variables, $\overset{-}{X}$ and $\overset{-}{Y}$ are the mean values of the samples.

## 3. Results

### 3.1. Process Electrical Response of PEO of Zirconium Alloy

Figure 3 shows the electrical response of the PEO process for samples D1, D2, D3 and D4. Starting from 0 to 80 s, the samples appear to be primarily anodized. Then, a sharp peak in the current density appears due to the ignition of the spark microdischarges. The distance d affected the maximum value of the RMS current density. As the distance d increases, the peak grows. Further, from 120 to 300 s, the current density nonlinearly decreases because of the oxide film growth. After 300 s, the current reaches a steady value, and the growth of the coating slows down.

The RMS voltage throughout the PEO process gradually rises following the coating thickness [28] because the matrix of pores created by the action of microdischarges affects the coating conductivity and effective capacitance. This has an impact on both the voltage pulses’ fall transient shapes and the RMS value [29]. Even though the samples are of the same thickness, D4’s final RMS voltage is lower than that of D3 one. This can be explained by an increase in the coating electrical conductivity caused by defects produced by more powerful microdischarges.

### 3.2. Electric Field Modelling Results

Typical current density distribution around the anode sample is shown in Figure 4. Figure 5 shows the relationship between the current density and distance *d*. The mutual influence of the plates decreases as the distance *d* grows, and the current density tends to maintain constant.

### 3.3. Quantitative Evaluation of the PEO Kinetics for the Zr-1Nb Alloy

In Formula (1), we used *m* = 1 to simplify the calculations; this assumes that the crystallization is instantaneous. The concentration *c* of Zr in the oxide layer was calculated for ZrO_2_ tetragonal and monoclinic phases using the crystalline phase content shown in Table 3:$$c_{Zr}=\frac{X({0.885\rho }_{tetragonal}+0.115\rho_{monoclinic})}{M_{ZrO_{2}}}$$
where *X* is the atom portion of Zr in oxide ZrO_2_.

The z–nuclear charge of Zr in oxide ZrO_2_ is equal to 4.

The current density transient was interpolated at the stages of the spark and microarc discharges, as the majority of the surface-layer modification happened during these stages. Figure 6 displays the current density curves obtained from the experiment and from the modelling. The value of R^2^ over 0.98 shows that a good fit was achieved.

The parameters of the kinetic model are shown in Figure 7. The initial (maximum) current density *j*_0_ of the dissolution term is affected by the magnitude of the peak current *j_RMS_* (Figure 3b) and the value of the crystallization term. The 3D crystallization curve contributes less, the current density peak is greater, and *j*_0_ is higher. The current peak increases as the shielding effect gets smaller as *d* gets larger. The initial value of the crystallization curve also grows with an increase in *d* and *j* because stronger microdischarges enhance active crystallization. As a result, *j*_0_ depends on *d* nonlinearly. With increasing *d*, the 3D crystallization coefficient grows. An increase in the Joule heating temperature at greater *j*_0_ can lead to an increase in the mobility of metal ions. Since time constant τ lowers as *d* increases, the formation of the coating occurs faster as the PEO current density gets higher.

### 3.4. Thickness and Morphology of the PEO Coating as a Function of the Samples Arrangement

The surface morphologies of the coatings after 360 s of plasma electrolytic oxidation are shown in Figure 8. The pore area distribution is displayed in Figure 9. The corresponding cross-sections are shown in Figure 10. The coating morphology parameters are presented in Figure 11. The coatings have a compact inner layer with a thickness 1.0–1.5 µm and a porous outer layer. The analysis of morphology of the surface (Figure 8) of the coatings shows that with an increase in the current density *j* on the surface of the coating, the area of homogeneous regions on the surface without pores increases (sample arrangement D1 to D4). To quantify this effect, the percentage of S_1_ (surface area without pores) to the overall surface area was calculated using the SEM micrographs. In addition, parameter S_2_ (the average area of one pore from a sample of pores more than 0.5 μm^2^) was also calculated. The results demonstrate that, on average, the pores grow bigger with an increase in *j*. This is supported by the pore area distribution. The parameters of the roughness are the arithmetic mean deviation of the assessed profile Ra and the maximum height of the profile Rz increases with growth of *h*. The differences in S_1_ are confirmed by the roughness parameters obtained by the profilometer: RPc (the number of profile elements per 1 cm of length that are above the set limit and immediately after that below the set limit) and RSm (mean spacing of profile elements). S_1_ changes in direct proportion to the RSm and inversely with RPc.

### 3.5. Thickness and Morphology of the PEO Coating as a Function of Time

Figure 12 shows SEM micrographs of the surface of samples D1 and D4 after the PEO during *t* = 60 s and *t* = 120 s; the corresponding pore area distribution is shown in Figure 13. The coating surface morphology after *t* = 360 s PEO is shown in Figure 8a,d and described in the previous section. The voltage pulse magnitude increased from 0 to 170 V during PEO time *t* = 60 s. This duration covers the electrochemical anodization stage. The black dots visible on the SEM micrograph in Figure 12a,b can be related to the anodic dissolution of the alloy. Sample D1 has lower coating thickness *h* and a higher area subjected to the anodic dissolution of S_3_ compared to sample D4. The voltage pulse magnitude reached 480 V during the treatment time of *t* = 120 s. Spark microdischarges caused the porous surface (Figure 12c,d) to form. For samples D1 and D4, the coating thickness and porosity *p*-values are statistically equal; this is consistent with the similar pore area distributions (Figure 13).

Due to stronger microdischarges that are developed throughout the treatment time of *t* = 360 s, the pores on the surface became bigger (Figure 8a,d). Sample D1 coating thickness grew by 14 ± 4% compared to the thickness gained by 120 s, and sample D4 coating increased by 36 ± 3%. Figure 14 displays a comparative coating thickness plot.

### 3.6. Phase Composition of the PEO Coatings as a Function of the Sample Arrangement

The XRD diffractograms of the coated Zr-1Nb alloy samples are displayed on Figure 15. Phase analysis of the coating formed during 6 min of the treatment shows that it mainly consists of the tetragonal *t*-ZrO_2_ phase with a small share of the monoclinic *m*-ZrO_2_ (Table 3). Tetragonal *t*-ZrO_2_ is a high-temperature phase that usually requires stabilization either with calcium, magnesium or yttrium oxides [30,31]. The diffraction peaks that belong to the substrate also appear in the diffractograms of the coated samples; however, their height is reduced because the X-ray electromagnetic wave when propagating in a solid dissipates following an exponential tendency characterized by a linear absorption coefficient of the material [32]. Therefore, the intensity of the X-rays diffracted in the substrate decreases due to the shielding effect of the coating.

**Table 3 materials-16-03377-t003:** PEO coating crystalline phase content (wt%) evaluated through the X-ray diffractometry.

Sample Code	Crystalline Phase Content in the PEO Coating (wt%)	
	t-ZrO*_2_*	m-ZrO*_2_*
D1	87 ± 2	13 ± 2
D2	87 ± 2	13 ± 2
D3	90 ± 2	10 ± 2
D4	91 ± 2	9 ± 2

### 3.7. Corrosion and Wear Resistance of the PEO-Coating

Figure 16 shows the polarization curves for the samples after the PEO at various sample arrangements. The corrosion parameters are combined in Figure 17. Analysis shows that the corrosion parameters for samples D1 and D2 are very close in the values. Sample D3 is characterized by a lower corrosion current *I_corr_* and a higher polarization resistance *R_p_*.

Figure 18a shows a comparison of the coefficient of friction (COF) in pin-on-disk wear tests on the PEO coated samples. All the samples exhibit similar curves; soon after the test started, the COF raises to 0.291 ± 0.016 for a while, then falls to 0.270 ± 0.015 and stays almost constant until the coating is fully worn out. After the coating is worn out, the COF rapidly increases for a short peak, and then stabilizes at a level of 0.413 ± 0.010 as the mutual wear of the Zr-1Nb substrate and Al_2_O_3_ ball occurs. The number of revolutions until failure *N* is shown in Figure 18b for comparison. The sample arrangement has no significant influence on the wear resistance parameter *N*; however, for sample D1, parameter N has the highest value.

## 4. Discussion

### 4.1. Influence of the Sample Spatial Arrangement on the Mechanism of the PEO Coating Growth on Zr-1Nb Alloy

The PEO process can be divided into three stages: anodizing, electrolytic–plasma processes with spark and with microarc microdischarges. During the anodizing stage, the operating voltage grows from zero to the breakdown voltage. The oxidation and dissolution processes compete during this stage. The first layer of the insulating oxide film is formed at this stage. Faraday’s law states that the thickness of the oxide layer grown by electrochemical deposition increases with greater current density. Experimental findings in Section 3.4 confirm this.

The first layer is destroyed in the following stage by the spark microdischarges. Moreover, in a sample with a thinner layer, microdischarges potentially happen sooner and at a lower voltage since the breakdown voltage is related to the thickness of the dielectric. Under the heat influence of the microdischarge, a portion of the metal melt is created, which is oxidized and fills (totally or partially generating pores) the cavity of the discharge channel [33]. Moreover, the oxide melt flows to the surface as a result of the gas buoyant forces. The layer expands in both directions (internal to the metal substrate and external to the electrolyte). When the operating voltage of 480 V is reached by 2 min, regardless of the sample arrangement, a coating with the same statistical thickness and porosity is produced. Lower current density is enough for spark microdischarges, which require less power for ongoing ignition. In addition, because the discharges ignited earlier, a sample with a lower current density can “catch up” in thickness to a sample with a higher current density. This indicates a significant contribution of plasma–chemical thermal processes in the formation of PEO coating at this stage in comparison with electrochemical processes.

The thickness difference becomes more noticeable at the following stage of the microarc discharges. Sample D1 coating thickness increased considerably less than that of sample D4 during the treatment period from 120 to 360 s (Figure 10). The coating thickness variation observed between D1 and D4 can be explained by a shielding effect of the nearby plates (distance 2.5 mm for D1). This effect decreases the local current density in the measurement spot more than by order of magnitude (Figure 5). As a result, the coefficient of diffusion D (Figure 7b) decreases twofold, thus decreasing the resultant coating thickness. Despite having a similar mechanism in the previous stage, the PEO stage with the microarc discharges varies due to the fact that breakdowns of the metal–electrolyte system happen less frequently but with more specific power. The growth of the coating “external” slows down more for the lower spatial current density sample D1 since the forming microdischarges lack the energy to transform the surface. However, some growth of the coating “internal” towards the metal is still possible.

With increasing coating thickness and roughness, the number of irregularities per square centimeter (RPc) decreases. The irregularities get bigger but become less numerous as the spatial current density rises. The oxide melt, which formed as a result of the heat action of a microdischarge, flows to the surface and makes “crater” pores there. As the discharge power grows, the pore average diameter (*S*_2_) increases. The micro defects are sealed by the melt spreading across the surface at greater spatial current densities and, consequently, create more potent and high-temperature microdischarges. This process results in the formation of smooth regions *S*_1_, the area of which is strongly correlated with *S*_2_ and RPc.

### 4.2. Correlation Analysis of the PEO Process Parameters and the Resultant Coating Properties on Zr-1Nb Alloy

Table 4 shows the correlation coefficients *r* of the factorial comparison of morphology, phase composition, corrosion properties and wear parameters. This table highlights well-known correlations (shown in blue), new uncovered correlations (shown in green), new correlations that statistically require additional studies (shown in gray) and indirect correlations due to the mutual influence of parameters on each other (shown in pink).

Correlation analysis showed that the distance between the plates *d* is correlated only with the current density *j*; therefore, no significant influence on the resultant coating properties is introduced by this important parameter. As the *E*corr corrosion potential becomes more negative, morphological parameters such *RS*m, *S*_1_ and *S*_2_ increase. Deep pore flaws make the surface less passivated, which explains why Ecorr and S_2_ relate to each other. Due to the proximity of the substrate to the depth of the pore where corrosion processes take place, the sample potential is changed in the direction of the alloy potential. Therefore, S2, *E*_corr_, *RS*m and S_1_ have a strong correlation. The decrease in *I*_corr_ with rising *S*_1_ can be explained through the idea that the area prone to corrosion reduces smooth areas with closed pores form on the surface. I_corr_ has a strong correlation with S_2_ since S_2_ correlates with S_1_. However, the appearance of even larger pores can lead to some deterioration of corrosion properties, as for sample D4.

The kinetic model coefficients are related to the coating morphological parameters. Considering that the current increases the temperature of the melt and improves the mobility of zirconium ions, the values of the coefficient *D* are correlated with the current density *j*. The coefficient *D* and parameter S1, which have a correlation with each other, are indirect measures of the melt volume. An increase in *D* expresses an increase in the rate of diffusion processes of three-dimensional crystallization of Zr oxide, which is confirmed by a decrease in the time constant τ. A sharp drop in current indicates a growth of the coating electrical resistance due to the rapid development of the oxide coating. A high correlation coefficient indicates the relationship between the time constant τ and the coating thickness and, accordingly, the surface roughness parameters Ra and Rz.

The 92% correlation coefficient between the number of revolutions before failure *N* and time constant τ suggests that the slower coating production may lead to a denser, more wear-resistant coating. It should be noted that the relationship exists between the crystalline phase composition of the coating and the parameters S_1_ and S_2_. It is assumed that an increase in the temperature of the microdischarges with increasing current density affects both the formation of the surface morphology and the increase of the amount of the high-temperature tetragonal phase t-ZrO_2_. This assumption explains the strong correlation of t-ZrO_2_ content with S_1_ and S_2_. The high correlation between t-ZrO_2_ and E_corr_ can be either explained by the correlation of E_corr_ with S1 and S2, or by the increased corrosion resistance of the tetragonal phase of zirconium oxide compared to that of the monoclinic phase. The variance of the t-ZrO_2_ parameter in different tests, on the other hand, is just a few percent and is not greatly beyond the error; therefore, these new correlations require additional proofs.

## 5. Conclusions

In this study, the influence of electrolyzer geometry in terms of the distance between two parallel plate samples subjected to plasma electrolytic oxidation on the resultant PEO coating properties and process kinetics was investigated. The analysis revealed that the PEO treatment can produce high-quality coatings in the investigated electrolyte system over a wide range of spatial distribution of current densities, which is necessary for processing objects with complex shapes. The spatial current density provides a notable impact on the surface morphology. The areas of the coated surface where the pores are sealed by the oxide melt expand due to the change in the power of microdischarges that occur with an increase in the spatial current density. The resistance to corrosion and wear are explained and correlated to the changes in the surface morphology, coating thickness and phase composition. The corrosion resistance reduces with increasing spatial current density, possibly as a result of larger longitudinal pores close to the metal-coating contact.

## Figures and Tables

**Figure 1 materials-16-03377-f001:**
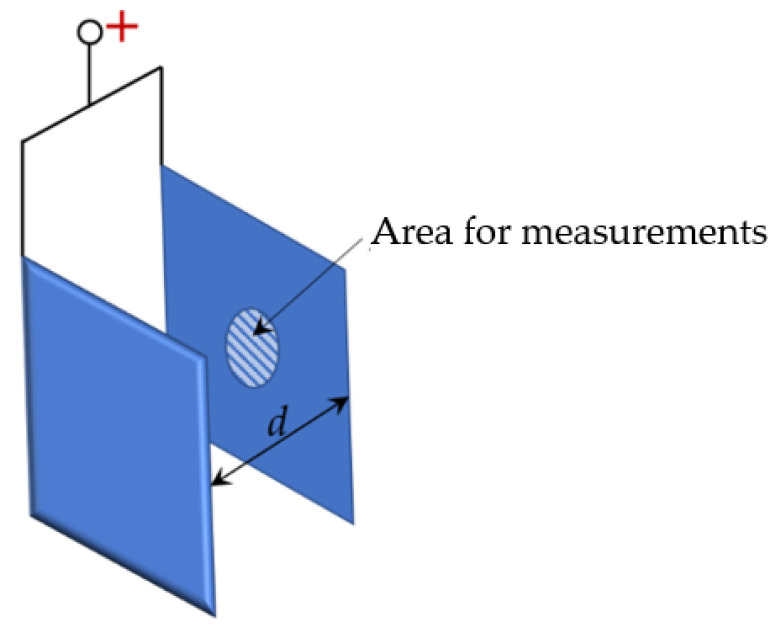
Illustration of two plates separated by distance *d*.

**Figure 2 materials-16-03377-f002:**
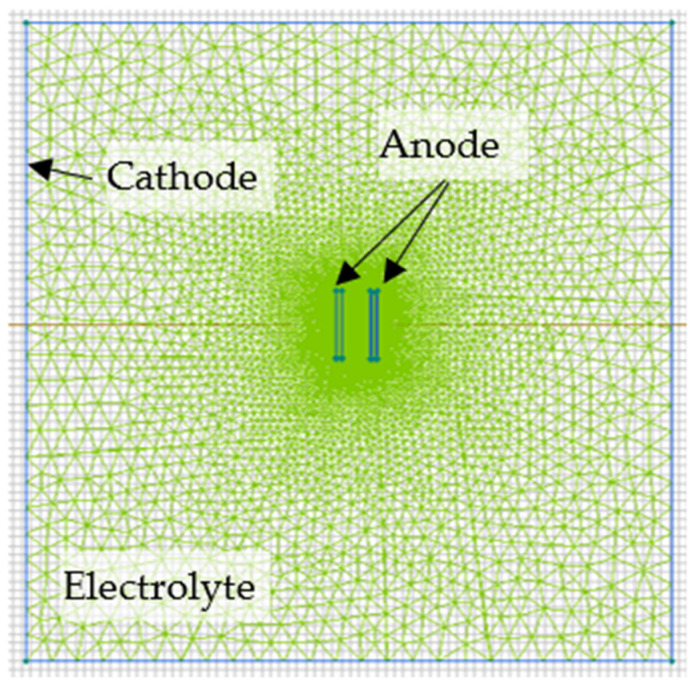
Schematic electrode arrangement and calculational grid in the electrolyte around the anode and cathode.

**Figure 3 materials-16-03377-f003:**
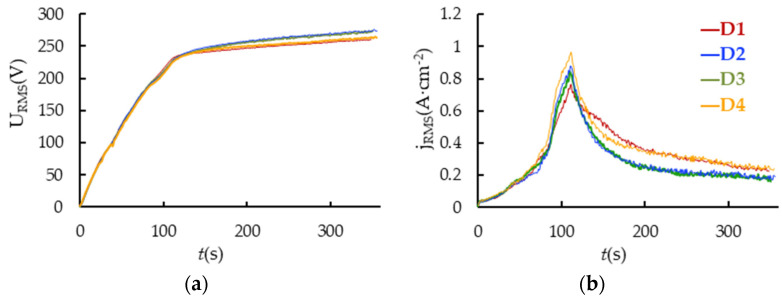
PEO process electrical response for the treatment of Zr1%Nb samples: (**a**) RMS voltage; (**b**) RMS current density.

**Figure 4 materials-16-03377-f004:**
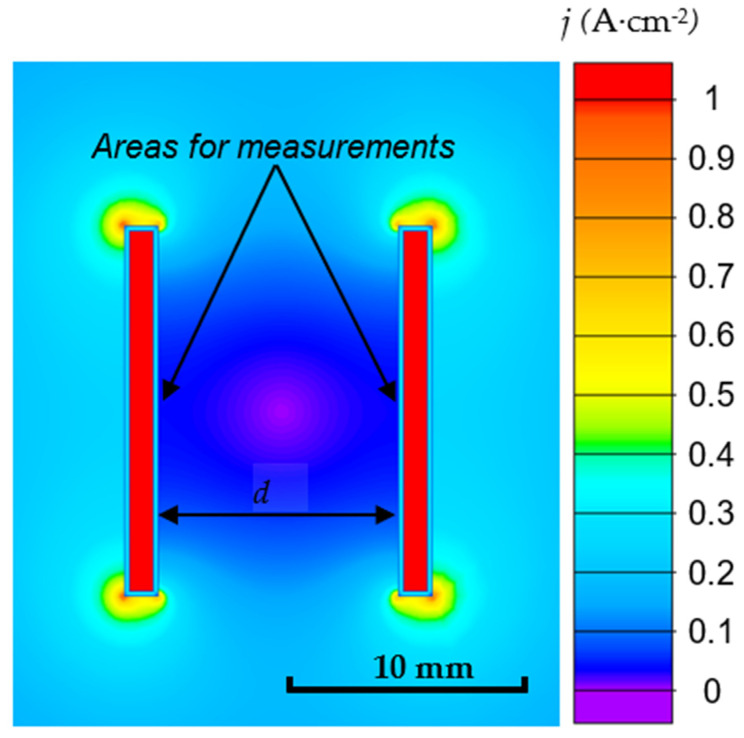
Distribution of the current density distribution in the electrolyte in the vicinity of the two anode plates (sample arrangement).

**Figure 5 materials-16-03377-f005:**
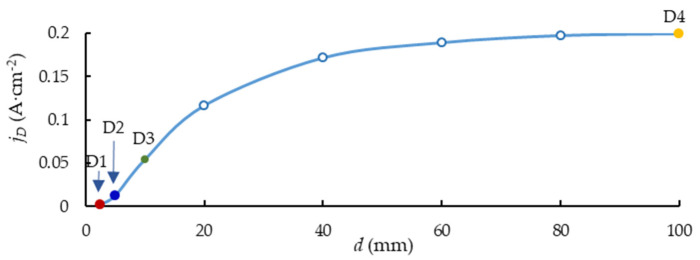
Current density at the areas of the measurement as a function of distance *d.*

**Figure 6 materials-16-03377-f006:**
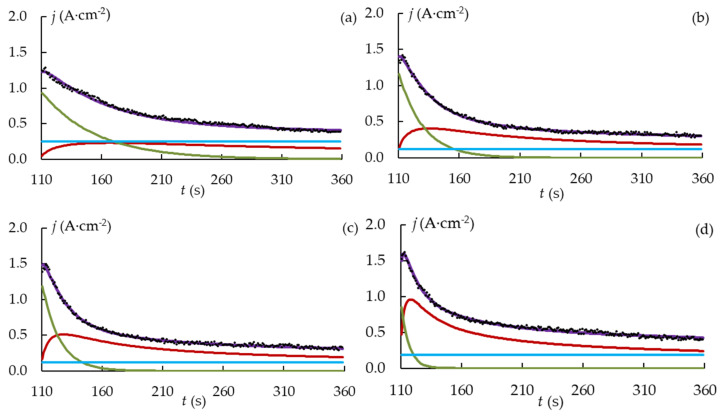
Current density curves obtained from the experiment and from the modelling of the PEO process of the samples D1 (**a**); D2 (**b**); D3 (**c**) and D4 (**d**). Modeled current density: **—** 3D-crystallization, **—** partial dissolution, **—** constant, **—** total.

**Figure 7 materials-16-03377-f007:**
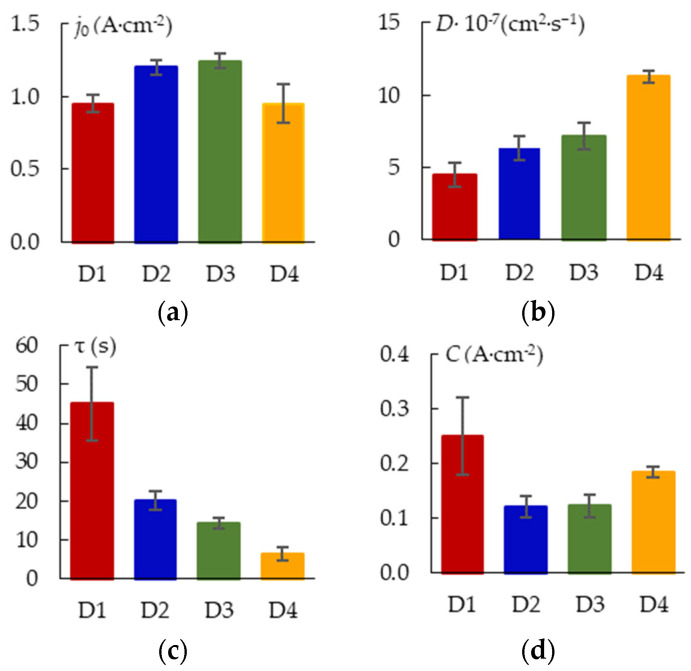
Parameters of the kinetic model: (**a**) j_0_; (**b**) D; (**c**) τ; (**d**) j_C._

**Figure 8 materials-16-03377-f008:**
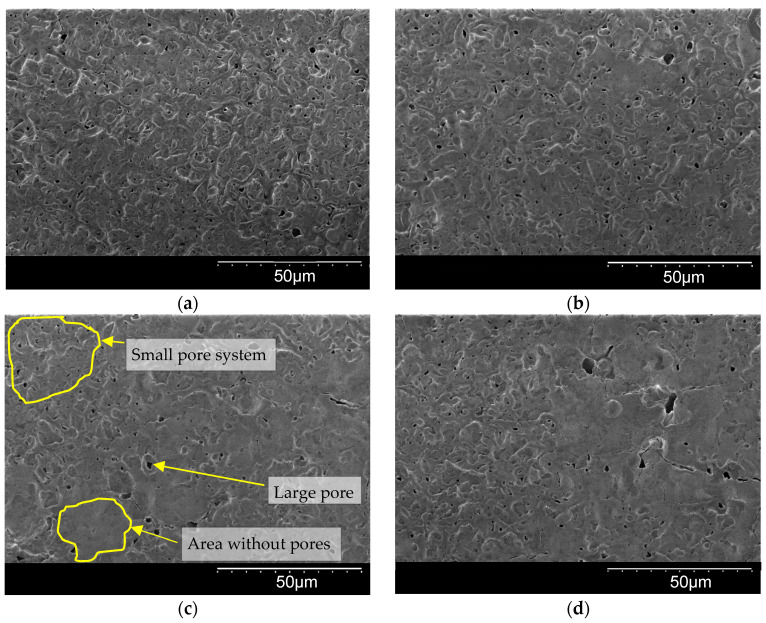
Microstructure of PEO coating on Zr-1%Nb for samples: (**a**) D1; (**b**) D2; (**c**) D3; (**d**) D4.

**Figure 9 materials-16-03377-f009:**
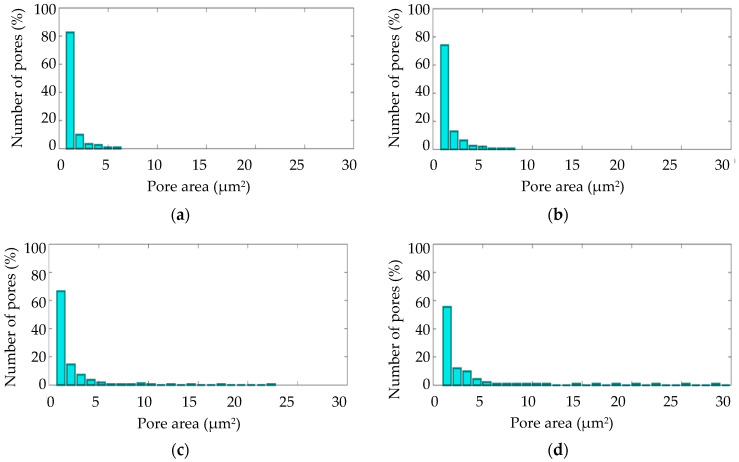
Pore area distributions for the PEO coating on Zr-1Nb for samples: (**a**) D1; (**b**) D2; (**c**) D3; (**d**) D4.

**Figure 10 materials-16-03377-f010:**
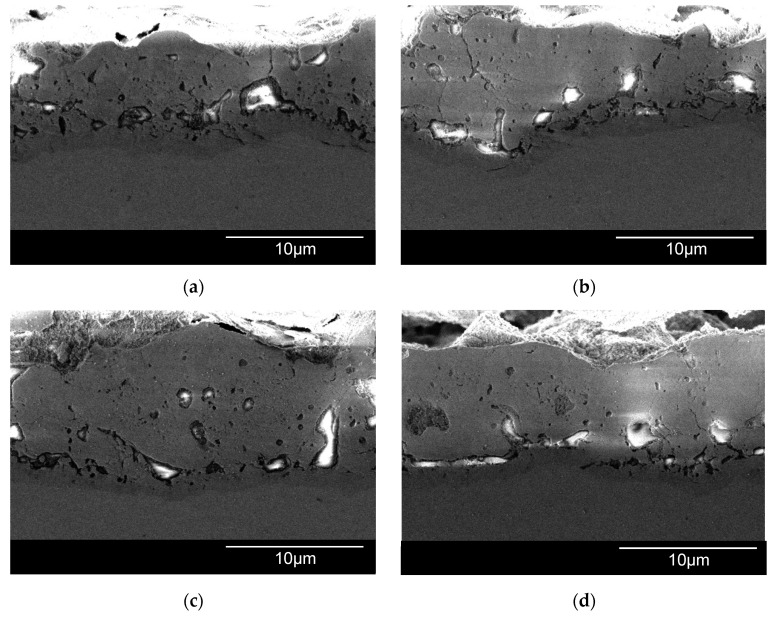
Cross-sections of the PEO coating on Zr-1Nb for samples: (**a**) D1; (**b**) D2; (**c**) D3; (**d**) D4.

**Figure 11 materials-16-03377-f011:**
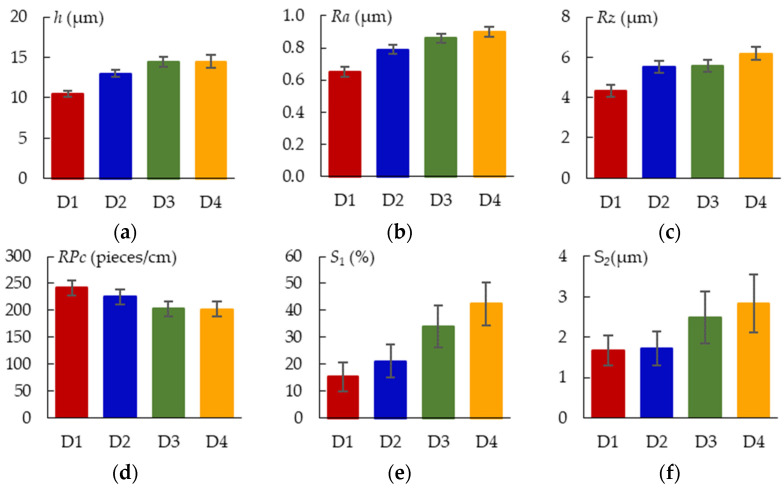
Parameters of the PEO coating morphology: (**a**) h; (**b**) Ra; (**c**) Rz; (**d**) RPc; (**e**) S_1_; (**f**) S_2_.

**Figure 12 materials-16-03377-f012:**
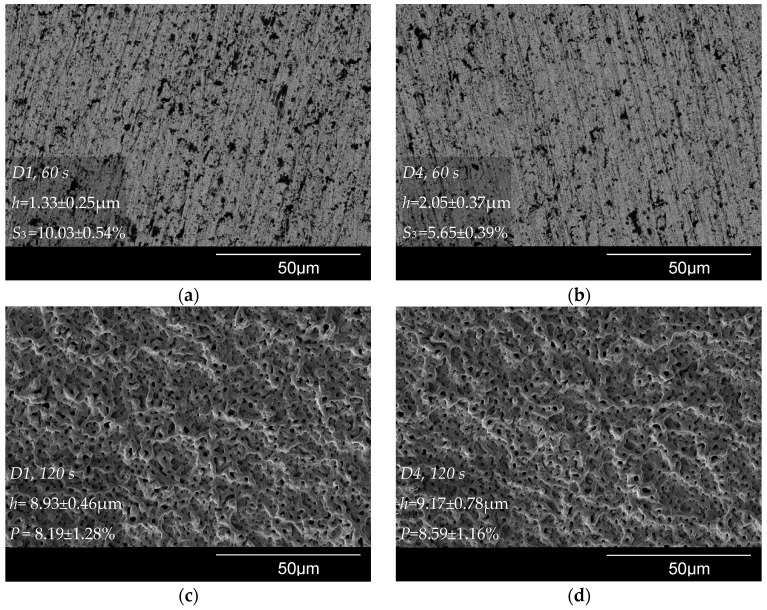
Surface morphologies of the PEO coatings for samples: (**a**) D1, *t* = 60 s; (**b**) D4, *t* = 60 s; (**c**) D1, *t* = 120 s; (**d**) D4, *t* = 120 s.

**Figure 13 materials-16-03377-f013:**
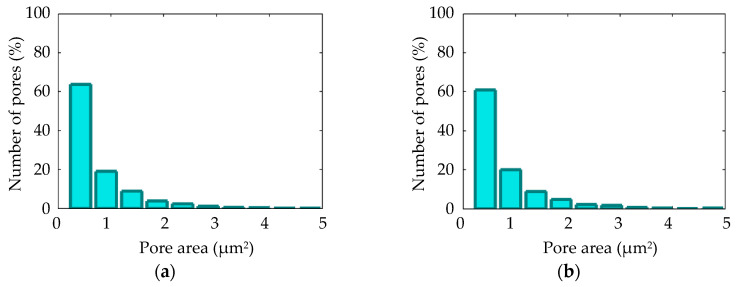
Pore area distributions for the PEO coating on Zr-1Nb for samples: (**a**) D1, *t* = 120 s; (**b**) D4, *t* = 120 s.

**Figure 14 materials-16-03377-f014:**
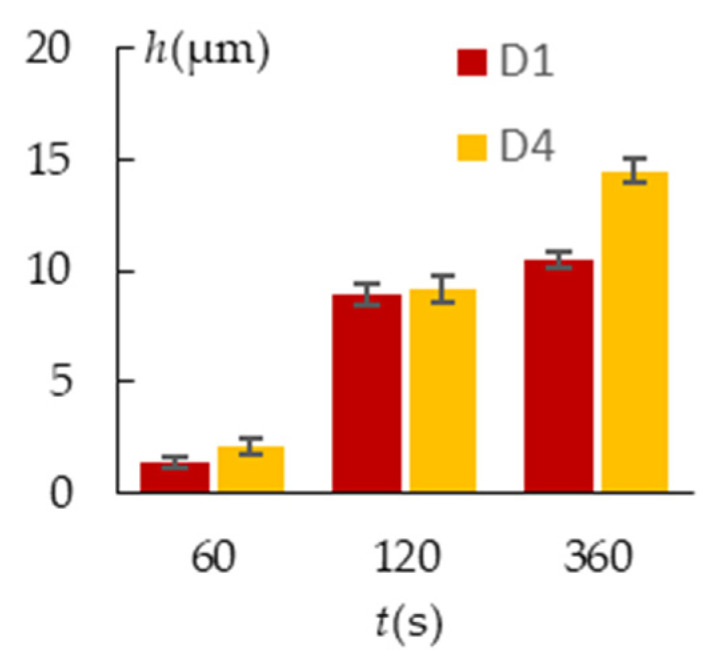
PEO coating thickness at different treatment times for sample arrangements D1 and D4.

**Figure 15 materials-16-03377-f015:**
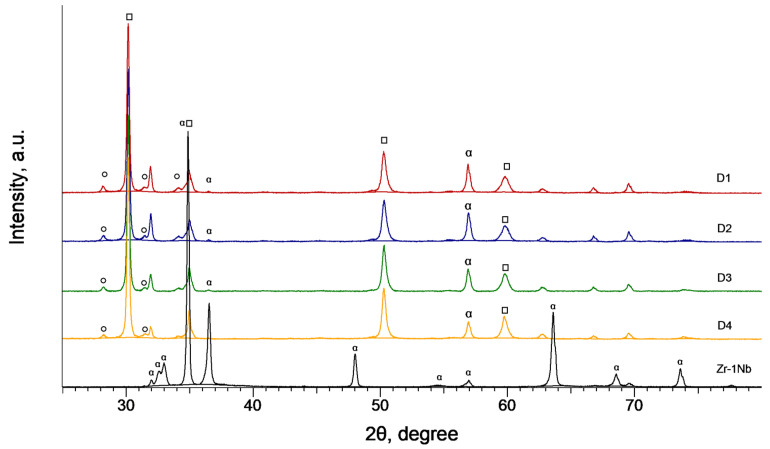
XRD diffractograms of the coatings formed on Zr-1Nb alloy after PEO treatment. ๐—m-ZrO_2_, □—t-ZrO_2_, α—α-Zr.

**Figure 16 materials-16-03377-f016:**
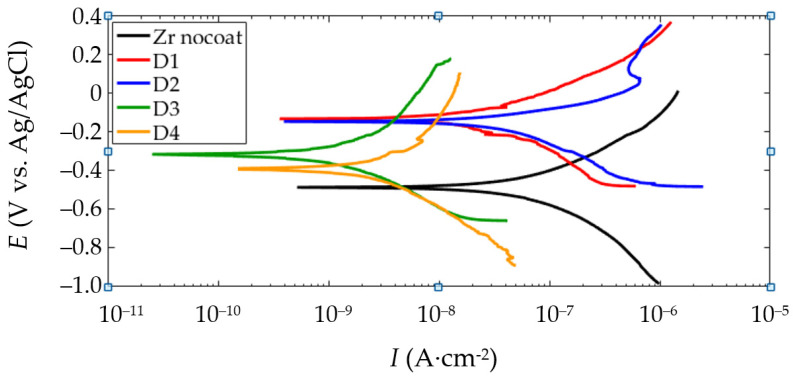
PDP plots of the electrochemical evaluation in Ringer’s solution for the PEO coated samples.

**Figure 17 materials-16-03377-f017:**
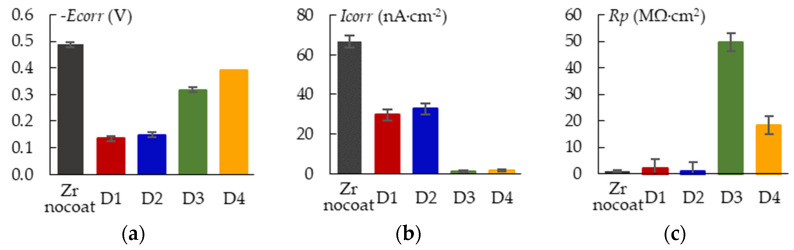
Results of potentiodynamic corrosion evaluation of the PEO coated samples in Ringer’s solution: (**a**) Ecorr; (**b**) Icorr; (**c**) Rp.

**Figure 18 materials-16-03377-f018:**
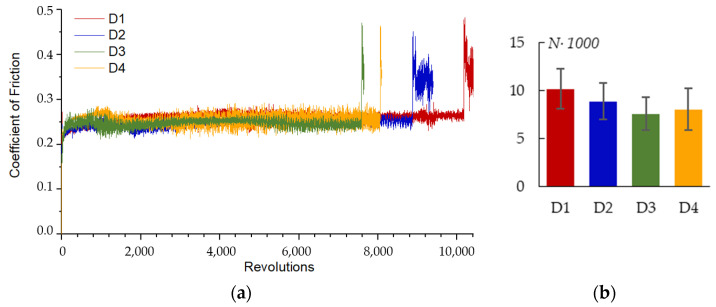
Coefficient of friction during pin-on-disk test for the PEO coated samples, 10 N load: (**a**) COF vs. revolutions; (**b**) number of revolutions until failure for different plate arrangements.

**Table 1 materials-16-03377-t001:** Sample arrangement in experiments D1–D4.

D1	D2	D3	D4
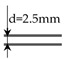	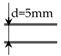	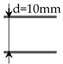	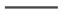

**Table 2 materials-16-03377-t002:** Chemical composition of Zr-1Nb alloy (weight%).

Nb	O	Hf	Fe	Ca	C	Ni	Cr	Si	Zr
1.10	0.10	0.05	0.05	0.03	0.02	0.02	0.02	0.02	Balance

**Table 4 materials-16-03377-t004:** Pearson correlation coefficient *r* (%) for PEO coating properties and kinetic model parameters.

	j	h	Ra	Rz	RPc	RSm	S1	S2	t-ZrO_2_	E_corr_	I_corr_	R_p_	N	j_0_	D	τ	C
d	98	55	65	70	−61	68	81	80	77	−80	−61	8	−43	−53	94	−65	11
j	100	67	76	77	−74	80	90	89	87	−89	−74	26	−57	−40	97	−74	0
h		100	99	95	−98	97	90	84	82	−84	−79	67	−98	40	80	−98	−71
Ra			100	97	−98	98	93	87	85	−87	−80	60	−95	29	87	−99	−64
Rz				100	−89	91	88	78	75	−79	−66	40	−86	23	89	−99	−63
RPc					100	−100	−95	−92	−92	92	90	−75	98	−29	−82	94	58
RSm						100	98	95	94	−95	−91	71	−95	21	87	−94	−52
S1							100	99	98	−99	−92	61	−86	−1	94	−90	−34
S2								100	100	−100	−96	67	−82	−8	89	−81	−22
t-ZrO_2_									100	−100	−98	70	−82	−7	87	−79	−22
E_corr_										100	96	−66	82	8	−89	82	23
I_corr_											100	−83	83	−4	−74	72	25
R_p_												100	−80	48	30	−51	−47
N													100	−50	−69	92	73
j_0_														100	−22	−31	−89
D															100	−86	−21
τ																100	68
							-well-known correlations
						-new uncovered correlations
						-new correlations that statistically require additional studies
						-indirect correlations, due to the mutual influence of parameters on each other

## Data Availability

Not applicable.

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
