# Peer review of "Influence of PEO Electrolyzer Geometry on Current Density Distribution and Resultant Coating Properties on Zr-1Nb Alloy"

_materials, 2023, doi:10.3390/ma16093377_

Round 1

Reviewer 1 Report

In this study, the influence of the electrolyzer geometry in terms of the distance between two parallel plate samples subjected to plasma electrolytic oxidation on the resultant PEO coating properties and process kinetics was investigated. This paper is very well written and very novel. I suggest accept the paper in its present form.

Author Response

Q: In this study, the influence of the electrolyzer geometry in terms of the distance between two parallel plate samples subjected to plasma electrolytic oxidation on the resultant PEO coating properties and process kinetics was investigated. This paper is very well written and very novel. I suggest accept the paper in its present form.

A: Thank you very much for your high evaluation of our work.

Reviewer 2 Report

The manuscript presents a comprehensive study of electrolytic oxidation of Zirconium alloy (Zr-1Nb) under different current density distributions. Experimentally, the authors used two Zr-1Nb plates as a collective anode and evaluate the effect of current density on morphology of the oxidized Zr, porosity, electrochemical stability, wear resistance. They used a number of experimental methods to perform structure analysis of the oxide and its mechanical and chemical stability. Also, they extracted the quantitative characteristics of oxidation kinetic parameters employing a kinetic model including metal dissolution, oxide crystallization and oxygen supply for numerical comparison. The quantification of process characteristics allows them to make the cross-correlation analysis between different quantities to reveal the dependencies. Both material and the oxidation process are of practical importance in fields of orthopedic implants and nuclear reactor components. The manuscript is well-organized and written, with clear figures and 28 references. I can recommend publishing it in the present form.

Author Response

Q: The manuscript presents a comprehensive study of electrolytic oxidation of Zirconium alloy (Zr-1Nb) under different current density distributions. Experimentally, the authors used two Zr-1Nb plates as a collective anode and evaluate the effect of current density on morphology of the oxidized Zr, porosity, electrochemical stability, wear resistance. They used a number of experimental methods to perform structure analysis of the oxide and its mechanical and chemical stability. Also, they extracted the quantitative characteristics of oxidation kinetic parameters employing a kinetic model including metal dissolution, oxide crystallization and oxygen supply for numerical comparison. The quantification of process characteristics allows them to make the cross-correlation analysis between different quantities to reveal the dependencies. Both material and the oxidation process are of practical importance in fields of orthopedic implants and nuclear reactor components. The manuscript is well-organized and written, with clear figures and 28 references. I can recommend publishing it in the present form.

A: Thank you very much for your high evaluation of our work.

Reviewer 3 Report

There are many mistakes and this paper is very disordered. They should be corrected. The discussion is not clear and it should be rewritten. Unfortunately, this paper is not suitable for publication.

Author Response

Q: There are many mistakes and this paper is very disordered. They should be corrected. The discussion is not clear and it should be rewritten. Unfortunately, this paper is not suitable for publication.

A: Thank you very much for your time devoted to our manuscript. Your brief and negative comment cannot be practically elaborated, as the mistakes, if any, are not pointed out. However, in general, we tried to follow your suggestion and improved the discussion.

Reviewer 4 Report

Gunderov's group reported the effect of various parameters during PEO coating. The manuscript is well organized, well written well and interesting. The manuscript can be accepted after minor revision. 

1. In XRD, many peaks related to Zr-1Nb alloy are not appeared after PEO treatment (Note: Some minor peaks were found after treatment). Author should provide necessary explanation for the disappearance such peaks.

2. The significant change in height was observed coating performed at 360 sec in the Sample D1 and D4. Author should explain this huge difference.      

Author Response

Q: Gunderov's group reported the effect of various parameters during PEO coating. The manuscript is well organized, well written well and interesting. The manuscript can be accepted after minor revision.

A: Thank you very much for your high evaluation of our work.

Q1. In XRD, many peaks related to Zr-1Nb alloy are not appeared after PEO treatment (Note: Some minor peaks were found after treatment). Author should provide necessary explanation for the disappearance such peaks.

A: The method of X-ray diffractometry has a limitation in the penetration depth of the x-rays into the surface layer. According to the solution of Helmholtz equation of an electromagnetic wave propagation in a solid, the following general form of the equation can be used:

I=I0*exp(-mu*L),

where I – intensity at depth L, I0 – incident intensity, mu – linear absorption coefficient of the material [https://doi.org/10.1002/jps.22202].

The thicker the coating is, the lower intensity of the diffraction lines coming from the substrate becomes.

Q2. The significant change in height was observed coating performed at 360 sec in the Sample D1 and D4. Author should explain this huge difference.     

A: The different coating thickness observed between D1 and D4 can be explained by a shielding effect of the nearby plates (distance 2.5 mm for D1). This effect decreases the local current density in the measurement spot more than by order of magnitude (Fig. 5). As a result, the coefficient of diffusion D (Fig. 7b) decreases two-fold; thus, decreasing the resultant coating thickness.

These comments have been introduced into the paper text in the appropriate places.

Reviewer 5 Report

Review for

materials-2333254

Influence of PEO electrolyzer geometry on current density distribution and resultant coating properties on Zr-1Nb alloy

The authors addressed the PEO processing parameters on the coating properties of Zr-1Nb alloy. The related research in this field is very limited in literature and this work presents a decent work and is significant to the research of Zr-1Nb materials. Albeit this work is in good shape (only needs minor revision), I think, nonetheless, that the manuscript could be improved if the authors could address the comments and recommendations I listed below.

In the introduction part, you should add more background information about the practical application of Zr-1Nb alloy. Like the nuclear industry. (https://doi.org/10.1016/j.jallcom.2017.01.356 ; https://doi.org/10.1007/s12666-022-02535-3)

PEO is a very common method to increase corrosion resistance. The more corrosion-related paper should be added. The following 2 papers may add to your work: (https://doi.org/10.1016/j.colsurfa.2023.131283; https://doi.org/10.1016/j.colsurfa.2022.130361)

You may need to include a statistic of pore distribution of size comparison, which is very common for PEO coating analysis. 

Significance :

. The scientific content of this paper is correct. 

. The technical quality of this paper is correct. 

Scientific soundness :

The subject addressed in this paper is relevant. 

Interest to the readers :

In my opinion, the method of this paper seems to be interesting for the readership of the journal.

Overall, this article is in good shape. You may consider my above suggestions. 

Author Response

Q: Influence of PEO electrolyzer geometry on current density distribution and resultant coating properties on Zr-1Nb alloy

The authors addressed the PEO processing parameters on the coating properties of Zr-1Nb alloy. The related research in this field is very limited in literature and this work presents a decent work and is significant to the research of Zr-1Nb materials. Albeit this work is in good shape (only needs minor revision), I think, nonetheless, that the manuscript could be improved if the authors could address the comments and recommendations I listed below.

Significance :

. The scientific content of this paper is correct.

. The technical quality of this paper is correct.

 Scientific soundness :

The subject addressed in this paper is relevant.

 Interest to the readers :

In my opinion, the method of this paper seems to be interesting for the readership of the journal.

 Overall, this article is in good shape. You may consider my above suggestions.  

A: Thank you very much for your high evaluation of our work.

Q1. In the introduction part, you should add more background information about the practical application of Zr-1Nb alloy. Like the nuclear industry. (https://doi.org/10.1016/j.jallcom.2017.01.356 ; https://doi.org/10.1007/s12666-022-02535-3).

A: We are grateful for the reviewer in reminding to make a focus on this important Zr application. We have included the references and their discussions as suggested.

Q2: PEO is a very common method to increase corrosion resistance. The more corrosion-related paper should be added. The following 2 papers may add to your work: (https://doi.org/10.1016/j.colsurfa.2023.131283; https://doi.org/10.1016/j.colsurfa.2022.130361)

A: We are grateful for the reviewer for this comment. However, the mentioned papers are devoted to sol-gel coatings on a Mg alloy. These are not focused on the PEO coatings of Zr, so it appears inappropriate to cite these papers. Nevertheless, we cited Zr PEO papers that concern the corrosion protection.

Q3: You may need to include a statistic of pore distribution of size comparison, which is very common for PEO coating analysis.

A: Following the reviewer’s recommendation, the statistics is included into Fig. 9 and 13.

Round 2

Reviewer 3 Report

a) In the abstract portion, authors have not demonstrated their achieved results (values), rather than focusing on general details, please rephrase to highlight your results as well to appeal to the readers.

b) In the introduction section, the problem statement does not clearly explain why the authors did this research and what was missing in the previous studies that the authors addressed here.

c) The comparison of this work with others reported in the literature must be added.

Author Response

The authors are grateful to the reviewer for valuable comments. The questions and answers are presented below.

Question a) In the abstract portion, authors have not demonstrated their achieved results (values), rather than focusing on general details, please rephrase to highlight your results as well to appeal to the readers.

Answer. The Abstract has been revised according to this suggestion.

Question b) In the introduction section, the problem statement does not clearly explain why the authors did this research and what was missing in the previous studies that the authors addressed here.

Answer. The Introduction has been revised according to this suggestion.

Question c) The comparison of this work with others reported in the literature must be added.

Answer. The Introduction has been revised according to this suggestion.